# Flow-induced periodic chiral structures in an achiral nematic liquid crystal

Qing Zhang [1,4] ✉, Weiqiang Wang [2,4], Shuang Zhou [3], Rui Zhang [2] ✉ & Irmgard Bischofberger [1] ✉

Supramolecular chirality typically originates from either chiral molecular building blocks or external chiral stimuli. Generating chirality in achiral systems in the absence of a chiral input, however, is non-trivial and necessitates spontaneous mirror symmetry breaking. Achiral nematic lyotropic chromonic liquid crystals have been reported to break mirror symmetry under strong surface or geometric constraints. Here we describe a previously unrecognised mechanism for creating chiral structures by subjecting the material to a pressure-driven flow in a microfluidic cell. The chirality arises from a periodic double-twist configuration of the liquid crystal and manifests as a striking stripe pattern. We show that the mirror symmetry breaking is triggered at regions of flow-induced biaxial-splay configurations of the director field, which are unstable to small perturbations and evolve into lower energy structures. The simplicity of this unique pathway to mirror symmetry breaking can shed light on the requirements for forming macroscopic chiral structures.

Chirality, or the absence of mirror symmetry, is ubiquitous in living systems, from DNA to the placement of organs in mammals[1-3]. Chiral objects in chemistry and materials science have revolutionised chemical catalysis[4,5], optical sensors[6,7], and metamaterial design[8-10]. There are two common ways how supramolecular chiral structures emerge. They can either be induced by a chiral input which in turn generates a chiral output, or they are composed of molecular building blocks that are themselves chiral[2,11]. By contrast, the emergence of chirality in centrosymmetric systems is much less common, and it requires spontaneous mirror symmetry breaking[12,13]. Elucidating the routes to induce mirror symmetry breaking in achiral molecular assemblies can guide technological advances exploiting chirality[14,15].

Liquid crystals (LCs) are materials composed of anisotropic mesogens. Achiral LCs of specific molecular shape are an example of a material that can form chiral structures[13,16-18]; the molecular bow shape of bend-core liquid crystals in the smectic phase, for instance, introduces chirality through an intralayer polar orientational ordering combined with a collective tilt of the smectic planes[19]. Hydrodynamic instabilities induced by oscillatory shear with a zero net flow can trigger chiral Williams domains in nematic thermotropic liquid

crystals[20,21]. Mirror symmetry breaking has also been shown to emerge in rod-shaped liquid crystals in the nematic phase when the material is confined to a specific spatial confinement that can be imposed by curved or inclined surfaces, or by hybrid surface anchoring conditions[13,22,23]. The mirror symmetry breaking induced by spatial confinement is particularly prevalent in nematic lyotropic chromonic liquid crystals (LCLCs)[24-29], aqueous dispersions of disk-shaped molecules that self-assemble into cylindrical aggregates. Over a range of temperatures and concentrations, LCLC solutions exhibit a nematic phase[30,31]. Due to the large aspect ratio of the aggregates, nematic LCLCs have a large elastic anisotropy: The twist Frank elastic constant, $K_2$, is an order of magnitude lower than the Frank elastic constants of splay, $K_1$, and bend, $K_3$[30,32]. If nematic LCLCs are forced to adapt to a curved surface imposed by spatial confinement, instead of relieving these deformations through splay and bend modes, they do so through a twist deformation that minimises the elastic free energy. Such a twist deformation is a pivotal element in forming chiral helices[24,26]. In addition to the Frank elastic constants of splay, twist and bend, the saddle-splay Frank elastic constant, $K_{24}$, also plays an essential role in triggering and stabilising chiral structures through

[1]Department of Mechanical Engineering, Massachusetts Institute of Technology, Cambridge, MA 02139, USA. [2]Department of Physics, Hong Kong University of Science and Technology, Hong Kong, China. [3]Department of Physics, University of Massachusetts Amherst, Amherst, MA 01003, USA. [4]These authors contributed equally: Qing Zhang, Weiqiang Wang. ✉e-mail: zqing@mit.edu; ruizhang@ust.hk; irmgard@mit.edu

lowering the elastic energy in cylindrical and toroidal geometries[29,33,34]. Exploiting the ease with which twist deformations occur and the non-negligible saddle-splay elasticity that stabilises chiral structures, pro-grammed surface anchoring conditions have been developed to con-trol chiral structures in achiral nematic LCLC solutions[35–37]. To date, structural chirality in achiral nematic LCLCs induced by anisotropic elasticities and confined boundary conditions has been reported exclusively in the static state at rest, where an imposed curvature or a pre-patterned surface is necessary for the emergence of chirality[13].

In this study, we discuss our discovery of flow-induced mirror symmetry breaking in nematic LCLC solutions in the absence of curved or patterned surfaces. We reveal the emergence of a distinct macro-scopic chiral structure when the material is flowing in a microfluidic cell; a periodic double-twist configuration that leads to a mesmerising stripe pattern perpendicular to the flow direction. We show that the periodicity of the stripes is governed by the competition between the viscous torque and the bend elastic torque acting on the director, and can be tuned by varying the gap thickness of the microfluidic cell or the flow velocity.

We demonstrate that the mirror symmetry breaking results from (i) the flow-tumbling character of nematic LCLC solutions and (ii) an elastic instability of a specific flow-induced configuration of the director field facilitated by the low energetic cost of twist deforma-tions. A tumbling nematic experiences a non-zero viscous torque for any orientation of the director ($\alpha_2 \alpha_3 < 0$, where $\alpha_2$ and $\alpha_3$ are the Leslie viscosity coefficients[38]), which destabilises the director field in shear flow. This induces distinct director configurations, including a biaxial-splay configuration characterised by opposite directions of the splay deformation in two orthogonal planes. We show that this biaxial-splay configuration, for which the stability solution is dictated by the saddle-splay elasticity, is unstable and evolves towards a lower energy state of the director field; the periodic double-twist configuration that is selected due to the small twist Frank elastic constant of LCLC solu-tions. This path to macroscopic chirality is unique and exceptionally simple: all it requires is a pressure-driven flow. The structural chirality is here triggered by a dynamic process when an achiral nematic material is driven away from equilibrium and relaxes to a chiral lower energy state.

## Results and discussion

### Flow-induced periodic double-twist structures in achiral nematic liquid crystals

An aqueous solution of 13 wt% disodium cromoglycate (DSCG) is placed in a rectangular microfluidic cell of length $l = 55$ mm, width $w = 40$ mm, and gap thickness $b = 8–26$ µm at room temperature $T = 23.2 \pm 0.5$ °C. At this concentration and temperature, DSCG solu-tions form a nematic phase[30,31]. We probe the director alignment of the nematic DSCG solution using polarising optical microscopy. The sur-face anchoring is planar and in the direction of the cell length, which we denote as $x$-direction. At rest, when imaged under crossed polariser (placed along the $x$-direction) and analyser, the material thus appears black, as shown in Fig. 1a, b.

Remarkably, at a volumetric flow rate of $q = 0.25$ µl/min, a large-scale stripe pattern spontaneously emerges perpendicular to the flow direction, as shown in Fig. 1a (see also Supplementary Video 1). Using a PolScope (OpenPolScope) to quantify the gap-averaged retardance map, we find that the retardance is low in certain regions of the stripes, indicating a homeotropic alignment where the director is parallel to the cell thickness direction, as shown in Fig. 1c. Between these regions, the director rotates from being parallel to the $z$-direction with a polar angle $\theta \approx 0°$, to being more parallel to the $y$-direction with an azimuthal angle $\varphi \approx 90°$ (Fig. 1c, d). The director thus undergoes a periodic twist deformation in the flow direction, as schematically shown in Fig. 1a. We denote the low-retardance regions as regions I in Fig. 1c, and the regions in between as regions II (Fig. S1).

To probe the alignment in the gap thickness direction, we use fluorescence confocal polarising microscopy. A low fluorescence intensity $I$ indicates an alignment of the director in the $y$-direction, a high fluorescence intensity indicates an alignment of the director in either the flow direction ($x$-direction) or the gap thickness direction ($z$-

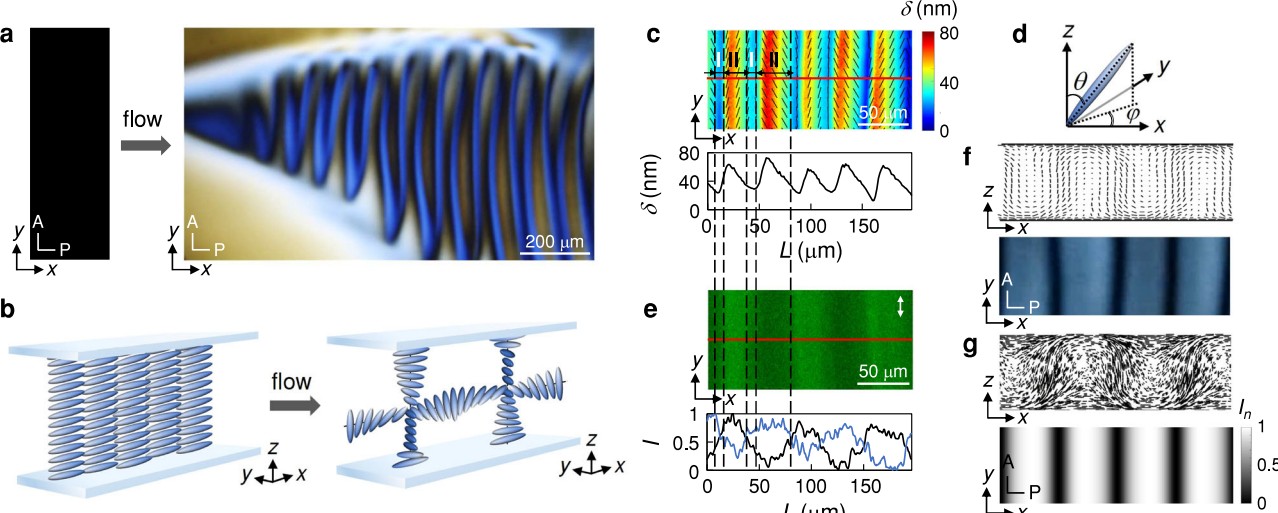

**Fig. 1 | Mirror symmetry breaking in flowing nematic LCLC solutions. a** Periodic stripe patterns emerge from uniformly aligned nematic LCLC solutions upon the onset of flow. **b** Schematics of the transition from a uniform planar alignment at rest to a periodic double-twist structure under flow. **c** Retardance map (upper panel), where the colour represents the optical retardance $\delta$ averaged across the gap thickness, and the black rods denote the orientation of the director averaged across the gap thickness and projected onto the $xy$-plane. Along the distance $L$ indicated as a red line, the retardance varies periodically (lower panel). The low-retardance regions are denoted as regions I, the regions in between as regions II. The gap thickness is $b = 14$ µm. **d** Director orientation. $\varphi$ is the azimuthal angle and $\theta$ is the polar angle. **e** Fluorescence confocal polarising microscopy image of the stripe pattern in the $xy$-plane imaged close to the bottom wall of the micro-fluidic cell (upper panel). The white arrow represents the polarisation of the probing beam. Along $L$, the normalised fluorescence intensity, $I$, measured close to the top wall (black line) is out of phase with that measured close to the bottom wall (blue line) (lower panel). **f** Schematic of the periodic double-twist config-uration (upper panel) and the corresponding stripe pattern (lower panel). **g** Map of the normalised light intensity under crossed polariser and analyser, $I_n$ (lower panel), determined from the simulated periodic double-twist director field (upper panel).

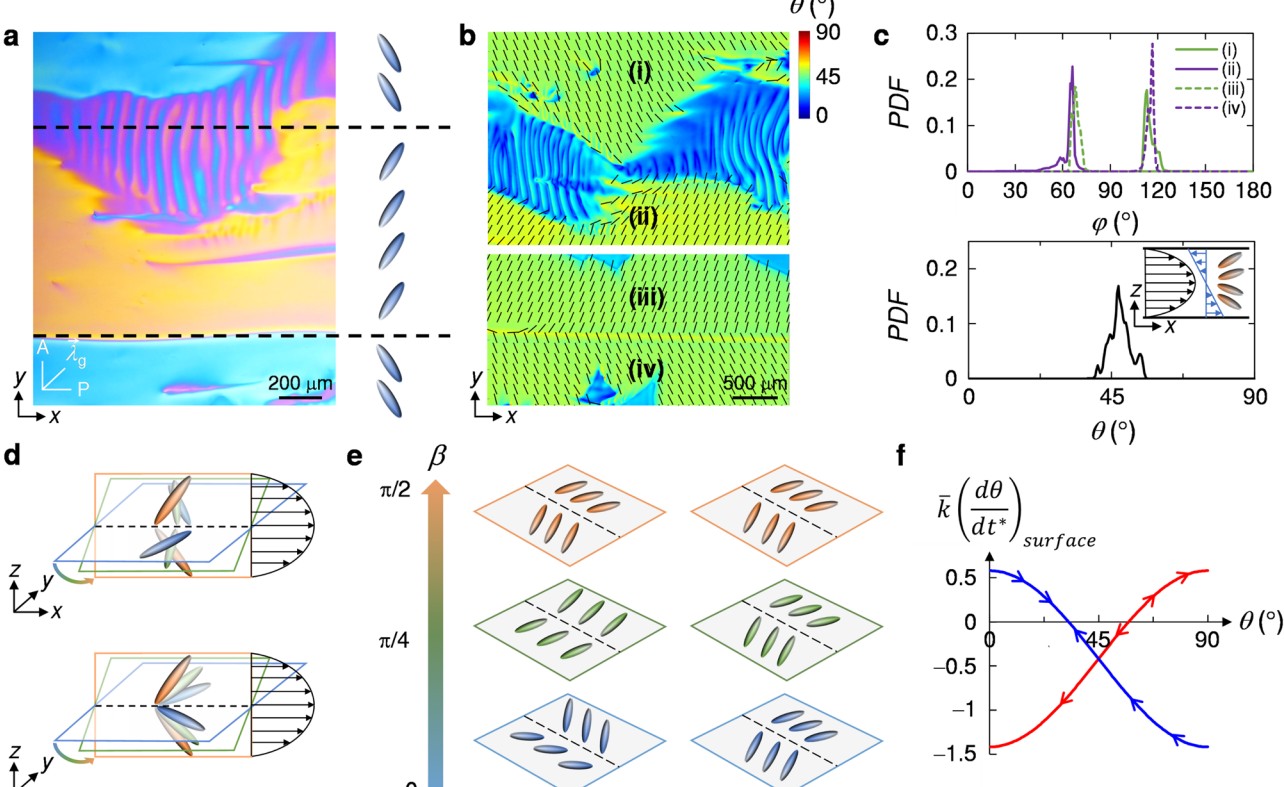

**Fig. 2 | Flow-induced double-splay and biaxial-splay configurations. a** The director field neighbouring the stripe pattern appears alternatively blue and yellow when imaged using crossed polarisers and a full-wave-plate optical compensator. The blue colour indicates that the director is almost perpendicular to the slow axis of the optical compensator, $\vec{\lambda}_g$, the yellow colour indicates that the director is almost parallel to $\vec{\lambda}_g$. **b** Map of the polar angle $\theta$ (colour bar) and the azimuthal angle $\varphi$ (black rods). The regions adjacent to the stripe pattern are denoted as (i) and (ii); the regions adjacent to the splay wall are denoted as (iii) and (iv). **c** Probability density function (PDF) of $\varphi$ (upper panel) and of $\theta$ (lower panel) in regions (i)–(iv). Inset: Schematic of the divergent splay deformation in the $xz$-plane induced by the pressure-driven flow. The black arrows represent the velocity profile, the blue arrows represent the shear rate profile. **d** Biaxial-splay configuration (upper panel) and double-splay configuration (lower panel). **e** With increasing rotation angle, $\beta$, the director field of the biaxial-splay configuration evolves from a convergent splay deformation in the $xy$-plane to a divergent splay deformation in the $xz$-plane (left). For the double-splay configuration, the director field adopts a divergent splay deformation for all $\beta$ (right). **f** Evolution of $\theta$ at the walls of the microfluidic cell ($z = 0$ and $z = b$) for double-splay (blue line) and biaxial-splay (red line) configurations. The arrows indicate the direction in which the director evolves upon a perturbation.

direction)[39]. In region II, the stripes exhibit alternatively high and low fluorescence intensity when measured at the bottom layer of the microfluidic cell. As we scan across the gap thickness towards the top layer of the microfluidic cell, the fluorescence intensity switches; dark regions become bright, and bright regions become dark, indicating an alternating twist deformation for adjacent stripes in the gap thickness direction, as shown in Fig. 1a, e.

The combination of the periodic twist deformation in the flow direction and the alternating twist deformation in the gap thickness direction results in a periodic double-twist structure, as schematically shown in Fig. 1f. To further verify that the periodic double-twist structure corresponds to the stripe pattern, we calculate the normalised light intensity averaged over the gap thickness from a simulated director field (see 'Methods'); the retardance map is in good agreement with the experimentally observed pattern (Fig. 1g). The periodic double-twist structure is remarkable in two aspects: (i) It is a chiral structure built by an achiral nematic liquid crystal, which involves a mirror symmetry breaking, and (ii) the structure possesses a well-defined characteristic period despite the absence of a pitch length in the achiral building blocks.

## Mechanism of mirror symmetry breaking

To reveal the mechanism of mirror symmetry breaking, we analyse the dynamics of the director field associated with the different elastic deformation modes induced by the flow. Looking at a larger section of

the cell through a crossed polariser, an analyser, and a full-wave-plate optical compensator with its slow axis, $\vec{\lambda}_g$, oriented at 45° to the polariser, we observe uniform regions surrounding the stripe pattern that appear alternatively blue and yellow. This observation, together with a quantification of the director field using a PolScope, demonstrates that the director rotates in opposite directions in the different uniform domains, as shown in Fig. 2a, b. This rotated director field results from the competition between the shear flow and the elastic deformation induced by the flow as the director resists deviation from the surface anchoring condition. At low shear rates, the tumbling DSCG director aligns perpendicular to the flow direction along the $y$-direction, adopting a log-rolling state that avoids the energetically costly splay and bend modes associated with a director deformation in the shear plane[38,40]. The log-rolling state, however, is inconsistent with the surface anchoring condition. This causes the director to rotate in the $xy$-plane (Fig. S2) adopting average azimuthal angles of $\varphi \approx 65°$ and $\varphi \approx 115°$, as shown in Fig. 2c. Given that the cylindrical DSCG aggregates are symmetric, left- and right-handed rotations are stochastically equivalent. At the boundary of two domains with oppositely rotated director fields, a splay deformation forms that is either open to the flow direction, denoted as divergent splay deformation or closed to the flow direction, denoted as convergent splay deformation. Remarkably, the stripe patterns only form at the boundaries with convergent splay deformation. By contrast, at the boundaries with divergent splay deformation a splay wall forms that appears as a sharp line in Fig. 2a[41].

Divergent and convergent splay deformations cost the same amount of energy; why is the divergent splay deformation stable at a splay wall but the convergent splay deformation unstable evolving into stripe patterns? The answer lies in the three-dimensional director field at the boundary of the domains. Indeed, in addition to the splay deformation in the $xy$-plane, the shear torques induced by the pressure-driven flow in the gap of the microfluidic cell induce a divergent splay deformation across the gap thickness, as shown in the inset of Fig. 2c. The deformation of the director field is reflected in the value of the gap-averaged polar angle $\theta \approx 45°$, which deviates from the initial planar alignment as a result of the tumbling property of nematic LCLC solutions[38,40,42,43], as shown in Fig. 2c. The combination of the convergent splay deformation in the $xy$-plane and the divergent splay deformation in the $xz$-plane induces a biaxial-splay configuration in which the director field undergoes a twist deformation when rotating about the symmetry line of the splay deformations with a rotation angle, $\beta$, as schematically shown in Fig. 2d, e. By contrast, the divergent splay deformations in both planes induce a double-splay configuration[44,45].

The observation that periodic double-twist structures are triggered at regions with biaxial-splay configuration, but not at regions with double-splay configuration, suggests different dynamics of the director field for these two configurations. Analysing the nematodynamic equations describing the dynamics of the director near the walls of the microfluidic cell (at $z = 0$ and $z = b$), we find that the two configurations have different stable solutions dictated by the saddle-splay elasticity. In the Frank-Oseen elastic energy density $f = (1/2)[K_1(\nabla \cdot \mathbf{n})^2 + K_2(\mathbf{n} \cdot \nabla \times \mathbf{n})^2 + K_3(\mathbf{n} \times \nabla \times \mathbf{n})^2 - K_{24}\nabla \cdot (\mathbf{n}(\nabla \cdot \mathbf{n}) + \mathbf{n} \times (\nabla \times \mathbf{n}))]$, where $\mathbf{n}$ is the director field and $K_1$, $K_2$, $K_3$ and $K_{24}$ are the splay, twist, bend and saddle-splay Frank elastic constants, the saddle-splay term enters the free energy only through the boundary conditions at the walls, given that it is a pure divergence term[46]. On a flat surface and when the surface anchoring condition or the bulk elastic energy are dominant, the saddle-splay term is typically neglected[29]. The polar surface anchoring strength of DSCG solutions, however, has been reported to be weak, on the order of $10^{-6}$ J/m$^2$ (ref. 47) (see Supplementary Information). When the flow induces a biaxial-splay or a double-splay configuration, the director near the walls of the microfluidic cell can thus deviate from the anchored state. This leads to spatial gradients of the director field in the orthogonal directions near the walls. As a consequence, the saddle-splay term plays here an important role. The director deformation at the walls of the microfluidic cell ultimately affects the director field in the bulk. To probe the stability of the biaxial-splay and double-splay configurations under perturbations, we consider the director field $\mathbf{n} = (\sin\theta\cos\varphi, \sin\theta\sin\varphi, \cos\theta)$ at the boundaries of the domains forming the biaxial-splay and double-splay regions. In steady-state flow, the polar angle $\theta$ and the azimuthal angle $\varphi$ are constant in the flow direction: $d\varphi/dx = 0$ and $d\theta/dx = 0$. Further assuming that $\varphi \approx 0°$ at the boundaries of domains with oppositely rotated director fields, the nondimensionalised nematodynamic equation at the top wall of the microfluidic cell reads (see Supplementary Information):

$$\bar{k}\left(\frac{d\theta}{dt^*}\right)_{surface} = -\frac{w}{b}\left(\frac{\partial\theta}{\partial z^*}\right)_{surface} + \frac{1}{2}\frac{K_{24}}{\bar{K}}\left(\cos^2\theta - \sin^2\theta\right)\frac{\partial\varphi}{\partial y^*}, \quad (1)$$

where $t^* = t/\tau$, $y^* = y/w$, and $z^* = z/b$, with $\tau$ the characteristic relaxation time of the director, and $w$ the characteristic width of the double-splay and biaxial-splay configurations. We define $\bar{k} = \frac{kw}{\bar{K}\tau}$, where $k$ is related to the rotational viscosity of the DSCG solution and $\bar{K} = (K_1 + K_3)/2$ is an average elastic constant. The surface gradient of $\theta$ in the $z$-direction, $(\partial\theta/\partial z^*)_{surface} = 0.836$, is solved numerically from the bulk nematodynamic equation (Figs. S3 and S4). For the director at the walls of the microfluidic cell where the flow velocity is zero because of no-slip boundary conditions, $(d\theta/dt)_{surface} = 0$ and Eq. (1) yields a solution for

the polar angle at the top wall $\theta_s \approx 57°$ for the biaxial-splay configuration and $\theta_s \approx 33°$ for the double-splay configuration. We now probe whether these are stable angles for the two configurations. For the double-splay configuration, the gradient of $\varphi$ in the $y$-direction, $\partial\varphi/\partial y^*$ is positive. When $\theta$ is perturbed to an angle smaller than $\theta_s$, the right-hand side of Eq. (1) becomes positive. The term $d\theta/dt^*$ is then positive and $\theta$ increases back to $\theta_s$. Inversely, when $\theta$ is perturbed to an angle larger than $\theta_s$, $d\theta/dt^* < 0$ and $\theta$ decreases back to $\theta_s$, as shown in Fig. 2f, where the arrows indicate the direction towards which the director relaxes after a perturbation. The director thus always returns to $\theta_s$, and the double-splay configuration is stabilised and forms a splay wall. In contrast, for the biaxial-splay configuration, $\partial\varphi/\partial y < 0$ and the director will not return to $\theta_s$ upon perturbation but evolve to either a homeotropic alignment ($\theta = 0°$) or a planar alignment ($\theta = 90°$). We show in the Supplementary Information (Figs. S3–S5) that due to the weak surface anchoring, the director evolves to a homeotropic alignment, which is unstable due to the mismatch with the surface anchoring condition and the non-zero viscous torque acting on the director field. There are two possible lower energy states out of the biaxial-splay configuration: a double-twist and a double-splay configuration. By analysing the Frank-Oseen elastic energy density for each configuration, we find that the double-twist configuration costs the least elastic free energy (see Supplementary Information and Fig. S6) and is therefore selected. We further rationalise the selection of the double-twist configuration by examining the elastic powers of splay, twist, and bend deformations in hybrid lattice-Boltzmann simulations. The elastic power of twist is indeed much larger than the elastic powers of splay and bend (see Supplementary Information and Fig. S7).

The biaxial-splay configuration that is the precursor for the periodic double-twist configuration emerges because of the imposed surface anchoring parallel to the flow direction. The stripe patterns do not form if we modify the surface anchoring condition so the director is planar and aligned in the $y$-direction; under these conditions, we observe a stable log-rolling state for a comparable range of flow velocities.

We determine the handedness of the periodic double-twist configuration using hybrid lattice-Boltzmann simulations. The twist deformations in the $x$- and $z$-directions exhibit the same handedness (see Fig. S8 and Table S1), which can be understood by considering that opposite-handedness configurations would necessitate energetically costly splay deformations in addition to the twist deformations. Across 21 independent simulations, we find an unbiased selection between left- and right-handed configurations, as shown in Table S1. This is a reflection of the DSCG aggregates' inherent achirality and the absence of external biases, underscoring the spontaneous nature of the mirror symmetry breaking. The elastic instability of the flow-induced biaxial-splay configuration of the director field provides a unique pathway to mirror symmetry breaking, where flow both triggers and stabilises the chiral periodic double-twist structure.

## Period of double-twist structures

The second remarkable characteristics of the stripe patterns is their periodicity. Periodic structures frequently appear in cholesteric liquid crystals where they result from the intrinsic pitch length of the material. In achiral nematic liquid crystals, however, they are rarely observed[22].

The period of the stripes that reflects the period of the double-twist structure, $p$, decreases with a power law with exponent $\approx -0.5$ with increasing velocity of the stripes, $V$, and increases with the gap thickness $b$, as shown in Fig. 3a. To rationalise these observations, we consider that the elastic deformation modes in the periodic double-twist configuration are predominantly bend and twist deformations. Given the small value of the twist Frank elastic constant, the bend mode dominates over the twist mode and competes with the viscous torque from the flow. We hypothesise that this competition sets the

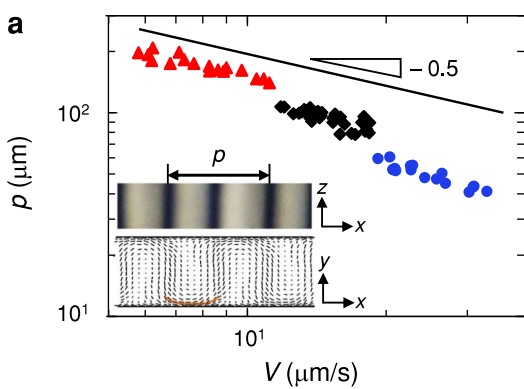

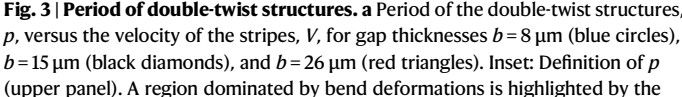

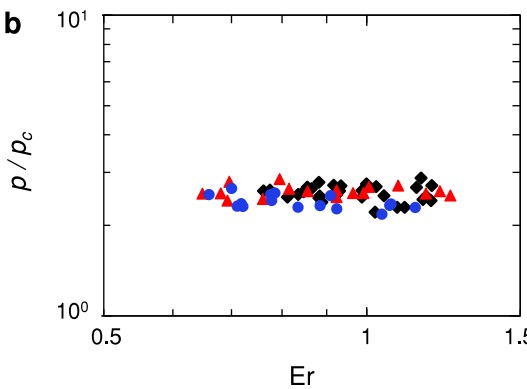

**Fig. 3 | Period of double-twist structures. a** Period of the double-twist structures, $p$, versus the velocity of the stripes, $V$, for gap thicknesses $b = 8\,\mu m$ (blue circles), $b = 15\,\mu m$ (black diamonds), and $b = 26\,\mu m$ (red triangles). Inset: Definition of $p$ (upper panel). A region dominated by bend deformations is highlighted by the orange line (lower panel). **b** $p$ normalised with $p_c$, the critical period denoting the competition between the bend elastic torque and the viscous torque from the flow, versus Ericksen number.

period of the structures. To probe this hypothesis, we consider the director field in the region of bend deformation, $\mathbf{n} = (\sin\theta, 0, \cos\theta)$. With the assumption that $\theta \approx 90°$ in this region, the nematodynamic equation reads (see Supplementary Information)[46]:

$$\alpha_3 \dot{\gamma}_{xz} = -K_3 \frac{\partial^2 \theta}{\partial x^2}, \quad (2)$$

where $\alpha_3$ is a Leslie viscosity coefficient and $\dot{\gamma}_{xz}$ is the shear rate in the $xz$-plane. A scaling analysis yields $\partial^2\theta/\partial x^2 \propto -1/(p_c/2)^2$ and $\dot{\gamma}_{xz} \propto V/b$ (see Supplementary Information), which provides a characteristic period,

$$p_c \propto 2\sqrt{\frac{K_3 b}{\alpha_3 V}}. \quad (3)$$

Normalising $p$ with $p_c$ indeed rescales all the data onto a master curve, as shown in Fig. 3b. We report $p/p_c$ versus the Ericksen number $Er = \eta_{eff} Vb/K_3$, where $\eta_{eff} \approx 0.43\,Pa\,s$ is the effective viscosity measured using a stress-controlled rheometer (AR-G2, TA Instruments, Fig. S9), and $K_3 \approx 10\,pN$ is the bend Frank elastic constant[38,48]. The stripes emerge for Ericksen numbers of order one, corroborating that the periodic double-twist structure emerges when the bend elastic torque competes with the viscous torque from the shear flow.

It is interesting to note that the configuration of the periodic double-twist structure is reminiscent of the periodic chiral structures that result from the Helfrich-Hurault elastic instability in cholesteric liquid crystals, where the period is set by the competition between the bend deformation and the pitch length-induced twist deformations[49]. For the achiral DSCG solutions that do not have an intrinsic pitch length, it is instead the viscous torque that resists the bend elastic deformation. Mirror symmetry breaking has further been observed in Williams domains forming under oscillatory shear, where the twist deformations are in the cell thickness direction[20,21]. In a related context, transient stripe patterns have been observed in LCLC solutions during the relaxation of the director field from a planar alignment to a homeotropic alignment imposed by the surface anchoring conditions[50]. Here, we show that steady flow can induce and stabilise chiral structures that extend periodically in the flow direction. The mirror symmetry breaking is facilitated by the tumbling character of the LCLC solution that triggers the three-dimensional director field to form a biaxial-splay configuration. This biaxial-splay configuration is unstable, as dictated by the saddle-splay elasticity, and evolves to a lower energy state, a chiral double-twist structure, as a consequence of the small twist

Frank elastic constant of LCLC solutions. A similar emergence of chirality might occur in other nematic materials that share the characteristics of tumbling and a small twist Frank elastic constant, for example in liquid crystal polymers and three-dimensional active nematics[30,51,52]. The ease with which the macroscopic chiral structures can be induced could be exploited to create programmable scaffolds for transmitting or detecting chirality at the molecular level[53-56].

## Methods
### Experimental methods
Nematic disodium cromoglycate (DSCG) (TCI America, purity > 98.0 %) solutions are prepared by dissolving DSCG in deionized water at 13.0 wt%[30]. The sample is heated to $T \approx 90°C$ where it is in the isotropic phase, which allows the DSCG to fully dissolve in water. The solution is then cooled to room temperature ($T = 23.2 \pm 0.5°C$) where it adopts the nematic phase[30,31].

The microfluidic cell consists of two glass plates separated by $8-26\,\mu m$ spacers (Specac, MY SPR RECT, OMNI). The width of the microfluidic cell is 40 mm, and the length is 50 mm. To avoid potential pulsatile effects inherent to syringe pump-controlled flows and to ascertain a uniform velocity profile at the inlet, we connect a reservoir of 25 mm in height, 40 mm in width, and 2–3 mm in interior thickness to one end of the microfluidic cell. Both cell surfaces are treated to introduce a uniform planar anchoring condition along the flow direction by following a protocol of surface rubbing, where the glass plates are rubbed along the cell length direction using a diamond particle paste with a particle diameter of $\approx 50\,nm$ (TechDiamondTools)[57].

To obtain the stripe patterns, the nematic DSCG solution is injected into the microfluidic cell through a 1 mm diameter hole at the top of the reservoir at controlled volumetric flow rates ranging between $q = 0.25-0.45\,\mu l/min$. The volumetric flow rate is set by a syringe pump (Harvard PHD 2000). We ensure the robustness of our findings by conducting tests utilising different microfluidic cells and syringe pumps, including cells without a reservoir. The stripe patterns consistently form for the same range of flow velocities, demonstrating that the stepping action of the syringe pump does not affect the results.

The flow field is observed through crossed polarisers and a static full-wave-plate optical compensator (560 nm) with the slow axis oriented at 45° to the polarisers, using an optical microscope (OMAX M837T) with objectives of magnification $M = 4\times$ and numerical aperture NA = 0.1, and $M = 10\times$ and NA = 0.25. This setup allows us to identify the director field averaged in the gap thickness direction. We further quantify the director field averaged in the gap thickness direction using a PolScope with $M = 5\times$ and NA = 0.15 (OpenPolScope).

We employ fluorescence confocal polarising microscopy (Leica SP5) with a water immersion objective of magnification $M = 25\times$ and numerical aperture NA = 0.95 to determine the director field of the stripe patterns in the gap thickness direction. We add fluorescent molecules (Acridine Orange, Biotium) at a concentration of 100 ppm to the DSCG solutions. In nematic DSCG solutions, the fluorescent molecules align with the orientation of the disk-like DSCG molecules that are perpendicular to the orientation of the director. The polarised probing beam excites the fluorescent molecules and induces fluorescence. The efficiency of excitation depends on the angle between the transition dipole of the fluorescent molecules and the polarisation of the probing beam[39]. A high fluorescence intensity indicates that the polarisation of the probing beam is parallel to the fluorescent molecules and thus perpendicular to the director of the DSCG aggregates. A low fluorescence intensity indicates that the polarisation of the probing beam is perpendicular to the fluorescent molecules and thus parallel to the director of the DSCG aggregates. In our experiments, the polarisation of the probing beam is perpendicular to the flow direction. When capturing the director field in the gap thickness direction, we scan eight layers from the top wall to the bottom wall of the microfluidic cell. During the scan, the stripe pattern moves with the flow. We therefore analyse the images in a Lagrangian framework in the frame of reference of the stripes. While taking the fluorescence images, we simultaneously capture images through crossed polariser and analyser. This allows us to trace the displacement of the stripes during the scanning process, and to correspondingly shift the region of interest for each layer to account for the motion of the stripes.

## Numerical methods

The numerical simulations employ the Leslie-Ericksen theory to account for the dynamics of the director in the regions where the director adopts specific splay configurations in a pressure-driven flow in a microfluidic cell of gap thickness $b$. The velocity field $\mathbf{u}$ and the unit-vector director field $\mathbf{n}$ are used to describe the nematodynamics of the nematic liquid crystal solution. The polar angle $\theta$ and the azimuthal angle $\varphi$ describe the director field $\mathbf{n} = (\sin\theta\cos\varphi, \sin\theta\sin\varphi, \cos\theta)$. The director field $\mathbf{n}$ in a steady-state flow is governed by the nematodynamic equation[46]:

$$\frac{1}{\gamma_1}\delta_{ij}^{\perp}h_j + W_{ik}n_k + \lambda\delta_{ij}^{\perp}A_{jk}n_k = 0, \qquad (4)$$

where $\gamma_1$ is the rotational viscosity, $\delta_{ij}^{\perp}$ is the transverse Kronecker delta, $W_{ik} = \frac{1}{2}\left(\frac{\partial u_i}{\partial x_k} - \frac{\partial u_k}{\partial x_i}\right)$ and $A_{jk} = \frac{1}{2}\left(\frac{\partial u_j}{\partial x_k} + \frac{\partial u_k}{\partial x_j}\right)$ are the antisymmetric and symmetric parts of the velocity gradients, and $\lambda = \frac{\alpha_2 + \alpha_3}{\alpha_2 - \alpha_3}$ with $\alpha_2$ and $\alpha_3$ the Leslie viscosity coefficients. $h_i = -\frac{\partial f}{\partial n_i} + \frac{\partial}{\partial x_j}\left(\frac{\partial f}{\partial(\partial n_i/\partial x_j)}\right)$, where $f = 1/2[K_1(\nabla\cdot\mathbf{n})^2 + K_2(\mathbf{n}\cdot\nabla\times\mathbf{n})^2 + K_3(\mathbf{n}\times\nabla\times\mathbf{n})^2 - K_{24}\nabla\cdot(\mathbf{n}(\nabla\cdot\mathbf{n}) + \mathbf{n}\times(\nabla\times\mathbf{n}))]$ is the Oseen-Frank elastic energy density. $K_1, K_2, K_3$ and $K_{24}$ are the splay, twist, bend and saddle-splay Frank elastic constants. Here, we focus on the configuration of the director field at the domain walls created by the flow-induced divergent or convergent splay configurations in the $xz$-plane, which allows us to simplify the nematodynamic equation into one-dimensional governing equations. We further note that $\varphi \approx 0°$ and $\frac{\partial\varphi}{\partial z} \approx 0$ at the domain walls. We approximate $K_1 \approx K_3 \approx \bar{K} \approx (K_1 + K_3)/2$, as $K_1$ and $K_3$ are of the same order of magnitude[30]. Eq. (4) then simplifies to

$$\bar{K}\frac{\partial^2\theta}{\partial z^2} = \left(\alpha_2\cos^2\theta - \alpha_3\sin^2\theta\right)\dot{\gamma}_{xz}, \qquad (5)$$

where $\dot{\gamma}_{xz} = \frac{\partial u_x}{\partial z}$ is the shear rate and $u_x$ is the velocity in the $x$-direction.

To describe the velocity field $\mathbf{u}$ in the $x$-direction, we employ a linear momentum equation[46]:

$$\eta_{eff}\frac{\partial^2 u_x}{\partial z^2} = -G, \qquad (6)$$

where $G$ is the pressure gradient in the $x$-direction. $\eta_{eff}$ is the effective viscosity, which is a function of the director field $\mathbf{n}$ and can be expressed as[46]

$$\eta_{eff} = \alpha_1\sin^2\theta\cos^2\theta\cos^2\varphi + \eta^b\sin^2\theta\cos^2\varphi + \eta^c\cos^2\theta + \frac{1}{2}\alpha_4\sin^2\theta\sin^2\varphi, \qquad (7)$$

where $\alpha_1$ and $\alpha_4$ are the Leslie viscosity coefficients and $\eta^b$ and $\eta^c$ are the Miesowicz viscosities[46,58].

We nondimensionalise Eqs. (5), (6) and (7) using $z = bz^*$, $u_x = \frac{Gb^2}{\alpha_2}u_x^*$, $\eta_{eff}^* = \frac{\eta_{eff}}{\alpha_2}$ and $\dot{\gamma}_{xz}^* = \frac{\partial u_x^*}{\partial z^*}$:

$$\frac{\partial^2\theta}{\partial z^{*2}} = \frac{Gb^3}{\bar{K}}\dot{\gamma}_{xz}^*\left(\cos^2\theta - \frac{\alpha_3}{\alpha_2}\sin^2\theta\right), \qquad (8)$$

$$\eta_{eff}^*\frac{\partial^2 u_x^*}{\partial z^{*2}} = -1, \qquad (9)$$

$$\eta_{eff}^* = \frac{\alpha_1}{\alpha_2}\sin^2\theta\cos^2\theta + \frac{\eta^b}{\alpha_2}\sin^2\theta + \frac{\eta^c}{\alpha_2}\cos^2\theta, \qquad (10)$$

where the Leslie viscosity coefficients are chosen to satisfy the tumbling character of flowing nematic lyotropic chromonic liquid crystal (LCLC) solutions; $\alpha_1 = -0.0181$ Pa s, $\alpha_2 = -0.1104$ Pa s, $\alpha_3 = 0.0011$ Pa s, $\eta^b = 0.0251$ Pa s and $\eta^c = 0.1355$ Pa s[30]. We numerically solve Eqs. (8) and (9) using the finite difference method, applying no-slip boundary conditions[59]. To account for the weak surface anchoring strength of LCLC solutions on our surface-treated glass plates[60], where the director can deviate from the initial surface anchoring condition in shear flow, we mimic finite surface anchoring conditions in our simulations that intrinsically have infinite surface anchoring conditions by assigning polar angles on the top and bottom walls of the microfluidic cell, $\theta_{b,top}$ and $\theta_{b,bottom}$, where $\theta_{b,bottom} = 180° - \theta_{b,top}$. We then test various $\theta_{b,top}$ and $\theta_{b,bottom}$ within the range of 45° to 90° to find the polar angles that correctly reflect the weak surface anchoring strength[61].

In Eq. (8), values are assigned to $Gb^3/\bar{K}$ based on experimental conditions and measurable variables. By scaling $G$ with $-\alpha_2\bar{V}/b^2$, we express $Gb^3/\bar{K} = -\alpha_2\bar{V}b/\bar{K} \equiv \mathrm{Er}_{av}$, where $\bar{V} = q/A$ is the average flow velocity, $q$ is the volumetric flow rate and $A$ is the cross-sectional area of the microfluidic cell. This yields Ericksen numbers $\mathrm{Er}_{av}$ in the range of 25–50 (corresponding to $\mathrm{Er} = \eta_{eff}Vb/K_3 = 0.65$–1.25 using the definition employed in the main manuscript).

To access the azimuthal angles at the domain walls, we employ the hybrid lattice-Boltzmann method where the director field of the nematic solution is described by the tensorial order parameter $\mathbf{Q}$ and the hydrodynamics by the velocity vector $\mathbf{u}$. For a uniaxial nematic liquid crystal, $\mathbf{Q} = S(\mathbf{nn} - \mathbf{I}/3)$, where $S$ is the scalar order parameter and $\mathbf{I}$ is the identity tensor. By defining the strain rate $\mathbf{D} = (\nabla\mathbf{u} + (\nabla\mathbf{u})^T)/2$ and the vorticity $\mathbf{\Omega} = (\nabla\mathbf{u} - (\nabla\mathbf{u})^T)/2$, we introduce an advection term $\mathbf{S} = (\xi\mathbf{D} + \mathbf{\Omega})\cdot(\mathbf{Q} + \frac{\mathbf{I}}{3}) + (\mathbf{Q} + \frac{\mathbf{I}}{3})\cdot(\xi\mathbf{D} - \mathbf{\Omega}) - 2\xi(\mathbf{Q} + \frac{\mathbf{I}}{3})(\mathbf{Q}:\nabla\mathbf{u})$, where $\xi$ is a constant that depends on the molecular details of the liquid crystal. We use $\xi = 0.6$ for our tumbling nematic LCLC solutions[61].

The governing equation of the **Q**-tensor, i.e., the Beris-Edwards equation, is[62]

$$\frac{\partial \mathbf{Q}}{\partial t} + \mathbf{u} \cdot \nabla \mathbf{Q} - \mathbf{S} = \Gamma \mathbf{H}, \tag{11}$$

where $\Gamma$ is related to the rotational viscosity via $\gamma_1 = 2S_0^2/\Gamma$ with $S_0$ the equilibrium scalar order parameter[63]. **H** is the molecular field defined as $\mathbf{H} = -(\frac{\delta F}{\delta \mathbf{Q}} - \frac{1}{3}\mathrm{Tr}(\frac{\delta F}{\delta \mathbf{Q}}))$ that drives the system towards thermodynamic equilibrium with a free energy functional $F = \int_{bulk} f_{LdG} dV + \int_{bulk} f_{elastic} dV + \int_{surface} f_{surface} dS$. The first term is the short-range Landau-de Gennes free energy density $f_{LdG} = \frac{A_0}{2}(1 - \frac{U}{3})\mathrm{Tr}(\mathbf{Q}^2) - \frac{A_0 U}{3}\mathrm{Tr}(\mathbf{Q}^3) + \frac{A_0 U}{4}(\mathrm{Tr}(\mathbf{Q}^2))^2$, where $A_0$ and $U$ are material constants[41]. The second term is the long-range elastic energy density $f_{elastic} = \frac{1}{2}L_1 Q_{ij,k} Q_{ij,k} + \frac{1}{2}L_2 Q_{jk,k} Q_{jl,l} + \frac{1}{2}L_3 Q_{ij} Q_{kl,i} Q_{kl,j} + \frac{1}{2}L_4 Q_{jk,l} Q_{jl,k}$, where $Q_{ij,k}$ denotes $\partial_k Q_{ij}$ using the Einstein summation convention[64]. The elastic constants $L_1$ to $L_4$ can be mapped onto the Frank elastic constants via[65]:

$$\begin{aligned}
L_1 &= \frac{1}{2S_0^2}\left[K_2 + \frac{1}{3}(K_3 - K_1)\right], \\
L_2 &= \frac{1}{S_0^2}(K_1 - K_{24}), \\
L_3 &= \frac{1}{2S_0^3}(K_3 - K_1), \\
L_4 &= \frac{1}{S_0^2}K_4.
\end{aligned} \tag{12}$$

The bulk elastic free energy density, $f_{bulk}$ is the sum of $f_{elastic}$ and $f_{LdG}$. The surface elastic free energy density $f_{surface} = \frac{1}{2}W(\mathbf{Q} - \mathbf{Q}_{surface})^2$ imposes a surface anchoring boundary condition to the **Q**-tensor by quadratically penalising any deviation of **Q** on a surface from the order parameter $\mathbf{Q}_{surface} \equiv S_0(\mathbf{n}_{surface}\mathbf{n}_{surface} - \mathbf{I}/3)$ imposed by the surface anchoring condition[59]. The parameter $W$ is the surface anchoring strength. We consider polar surface anchoring, $\mathbf{n}_{surface} = \mathbf{x}$, with **x** a unit director along the $x$-direction. The polar surface anchoring is weak with $W = 0.02$ (in simulation units).

The local fluid density $\rho$ and the velocity **u** are governed by the Navier-Stokes equation[66,67]

$$\rho\left(\frac{\partial}{\partial t} + \mathbf{u} \cdot \nabla\right)\mathbf{u} = \nabla \cdot \boldsymbol{\Pi} + G\mathbf{x}. \tag{13}$$

The viscoelastic properties of the nematic LC solution are lumped in the passive stress $\boldsymbol{\Pi}$ that is the sum of the viscous and elastic terms[59,63]:

$$\begin{aligned}
\boldsymbol{\Pi} =\ & 2\eta\mathbf{D} - P_0\mathbf{I} + 2\xi\left(\mathbf{Q} + \frac{\mathbf{I}}{3}\right)(\mathbf{Q}:\mathbf{H}) - \xi\mathbf{H}\cdot\left(\mathbf{Q} + \frac{\mathbf{I}}{3}\right) \\
& - \xi\left(\mathbf{Q} + \frac{\mathbf{I}}{3}\right)\cdot\mathbf{H} - \nabla\mathbf{Q}:\frac{\delta F}{\delta\nabla\mathbf{Q}} + \mathbf{Q}\cdot\mathbf{H} - \mathbf{H}\cdot\mathbf{Q},
\end{aligned} \tag{14}$$

where $\eta$ is the isotropic viscosity and $P_0$ is the isotropic bulk pressure. We consider a pressure-driven flow along the $x$-direction.

We employ the hybrid lattice-Boltzmann method to solve the coupled governing partial differential Eq. (11) and Eq. (13)[59,66,67]. The simulation box size is $[L_x, L_y, L_z] = [5, 51, 5]$, and we have periodic boundary conditions in the $x$- and $y$-directions. We choose the following parameters: $\eta = 1/3$, $\Gamma = 0.1$, $\xi = 0.6$ and $U = 3.5$, which results in $S_0 \simeq 0.62$. We further use $A_0 = 0.1$, $L_1 = 0.1$, $L_2 = 0$, $L_3 = 0.3247$, and $L_4 = 0.133$, corresponding to $K_1 = 3K_2 = \frac{1}{3}K_3 = K_{24}$ in Eq. (12) for flow-tumbling nematic LCLC solutions[68]. No-slip boundary conditions are imposed at the two walls of the microfluidic cell.

## Simulated director field of the stripe pattern

To reconstruct the three-dimensional director field of the stripe pattern, we use continuum simulations to generate a nematic director field that satisfies the experimentally observed director field, starting the simulation with an ansatz that is consistent with the twisted structure observed in the experiments. The structure is then stabilised by minimising the free energy of the system[64]. The equilibrated director field is processed to generate a crossed-polarised image[69], which agrees well with the experimental image supporting the validity of the reconstructed director field.

## Data availability

The authors declare that the data supporting the findings of this study are available within the text, including in the Methods section and Source Data files. Source data are provided in this paper.

## Code availability

The code used in this study is available on *Zenodo* under the accession code: https://doi.org/10.5281/zenodo.10155871.

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

## Acknowledgements

We thank A. Chen, Z. Wang and Z. Feng for assistance with the experiments and simulations. Q.Z. and I.B. acknowledge support from the National Science Foundation award number DMR-2245163. R.Z. and W.W. acknowledge support from the Hong Kong RGC grant no. 16300221 and no. 26302320. S.Z. acknowledges support from the National Science Foundation award number DMR-2239551 and the UMass Amherst startup fund.

## Author contributions

Q.Z. and I.B. designed the research. Q.Z. performed the experiments. Q.Z., W.W. and R.Z. performed theoretical analyses. W.W. and R.Z. performed the numerical simulations. Q.Z., W.W., S.Z., R.Z. and I.B. analysed the data and wrote the paper.

## Competing interests

The authors declare no competing interests.
