## [Peer Review File · Nature Communications]

REVIEWER COMMENTS

Reviewer #1 (Remarks to the Author):

The paper reports on the observation of creating chiral microstructures in pressure driven flow of achiral nematic complex fluid. The authors observe emergence of double twist organization with a characteristic length, which they show is determined by the balance between elastic and viscous forces. The experiments are well complemented with basic theoretical analysis. The paper is well organized and clearly written.

The authors show that the chiral structure emerges as a results of spontaneous symmetry breaking, which implies that both handed structures should be equally likely. Previously it was observed that in such systems often domains of opposite handed domains emerge; so my question is do the authors observe emergence of domains, and if not is there some fundamental reason why not? Overall, I think the work would benefit from some further elaboration and results on the exact mechanism(s) that control the spontaneous symmetry breaking and their consequences. Also, can these mechanism be biased to some handedness?

Finally, I think this is a relevant and well executed work which is appropriate and of relevance to be published in Nature Comm, upon addressing the comments above.

Reviewer #2 (Remarks to the Author):

The manuscript presents an experimental liquid-crystalline system which induces an chiral symmetry breaking in an achiral material with flow.

The material in question is a lyotropic chromonic liquid crystal - they are known to promote chiral symmetry breaking due to low twist elastic constant, which has been demonstrated in various static systems before. In flow, additional torques act on the liquid crystal constituents, leading to stabilization of structures that would otherwise relax to a static ground state.

The main contribution of the manuscript is demonstrating in experiment, the first flow realization of a chiral striped pattern in an achiral system. The pitch of the stripes can be regulated with flow velocity. The structure is fully backed by theoretical considerations, including stability analysis that

explains the formation of the striped pattern, and relations that collapse the measurements onto the same curve. I find the analytical explanations elegant and effective. The experiments are well documented and explore the system with different techniques to provide a full picture, including a fluorescent confocal imaging.

The research is of sufficient prominence and novelty to merit publication in suggested journal. It is of interest to readers in soft matter, optics and hydrodynamics. It extends the range of materials for which microfluidic response is explored, and provides a way of generating optically interesting patterns with minimal effort.

Before publication, the following should be clarified: it is not immediately clear from the text whether the chirality can alternate left and right-handed already within the same periodic pattern, or only in different realizations of the same pattern.

Additionally, chirality itself only comes to prominence in phenomena that can exploit and detect it. This should be discussed - for example, chirality of the system could be more convincingly proven by optical discrimination through different response to circularly polarized light.

Reviewer #3 (Remarks to the Author):

Zhang and coauthors observe interesting symmetry breaking effects in driven chromonic liquid crystals. They find that the combined large elastic anisotropy of chromonics and the pumped viscous effects create beautiful stripe patterns. Unfortunately there are several glaring issues with the work as reported here that should be addressed before further consideration, both experimental and simulation/analysis related.

The authors repeatedly state they used pressure control to drive the flow but the methods say they used a syringe pump—in this case, they used volume control (syringe pump) at fairly low flow rates to induce the flow. The stepping action of the syringe pump could cause issues. Would results look different with actual pressure control? The simulations seem to indicate they used pressure control. This could have substantial effects on the observed patterns.

Reference 44 does not really support the authors' claim regarding polar anchoring energy as it is mainly about azimuthal anchoring. (They use W_ϕ not W_θ) The true polar anchoring energy on a bare or abrasive treated glass surface is unknown but seems like it would be much higher than the authors indicate here at $1e-6$ – $1e-7$ J/m². Small tilts would induce large splay deformations which are especially energetically costly for the large aggregates in DSCG. This point is key to the discussion regarding saddle-splay and more clarification would help here. Do the authors see the same result on bare (not rubbed) glass? Glass treated for homeotropic alignment?

In general, the observations here remind of the Williams domain and flexo domain effects (which I believe may break mirror symmetry too) and other similar patterns that emerge in LC dynamic flows. It would be good for the authors to first revisit that deep literature. In that context, are the observations here that novel?

Minor points:

The authors have missed what seems like a low hanging fruit: compare energy densities in the system graphically. Elastic energies and stress energies would be revealing at a glance to the energy competition that creates the symmetry breaking. I suspect the biaxial-splay (or saddle splay concentrated) points would stand out, no?

And in the abstract, the terminology vertical to the flow direction is confusing.

Responses to reviewers' comments

Reviewer 1

The paper reports on the observation of creating chiral microstructures in pressure driven flow of achiral nematic complex fluid. The authors observe emergence of double twist organization with a characteristic length, which they show is determined by the balance between elastic and viscous forces. The experiments are well complemented with basic theoretical analysis. The paper is well organized and clearly written.

We very much appreciate the careful assessment of our manuscript.

The authors show that the chiral structure emerges as a result of spontaneous symmetry breaking, which implies that both handed structures should be equally likely. Previously it was observed that in such systems often domains of opposite handed domains emerge; so my question is do the authors observe emergence of domains, and if not is there some fundamental reason why not?

We thank the reviewer for the insightful question regarding the handedness of the domains. As it is difficult to identify the handedness of the periodic double-twist configuration in our experiments, we turn to numerical simulations using the hybrid lattice Boltzmann method to access the handedness of the stripe domains [1]. To closely mirror the properties of nematic DSCG solutions [2], we implement distinct elastic constants, emphasizing a small K_2 and a large K_{24} , and ensure a flow-tumbling character. With the liquid crystal solution confined between two plates (in the z -direction), and employing no-slip and finite planar surface anchoring conditions, periodic double-twist configurations emerge upon the onset of a pressure-driven flow and small perturbations, as shown in Fig. R1.1.

In all 21 independent simulations, we observe that the twist deformations in the x - and z -directions exhibit the same handedness (Fig. R1.1 and Table R1.1). This can be understood by considering that opposite-handedness configurations would necessitate energetically costly splay deformations in addition to the twist deformations present in both scenarios.

Right- and left-handed chiralities emerge with equal probability (10 left-handed and 11 right-handed configurations in the 21 independent simulations). We have added these results in both the manuscript and the Supplementary Information.

The texts now read:

Manuscript: *'We determine the handedness of the periodic double-twist configuration using hybrid lattice Boltzmann simulations. The twist deformations in the x - and z -directions exhibit the same handedness (see Fig. S7 and Table S1), which can be understood by considering that opposite-handedness configurations would necessitate energetically costly splay deformations in addition*

to the twist deformations. Across 21 independent simulations, we find an unbiased selection between left- and right-handed configurations, as shown in Table S1. This is a reflection of the DSCG aggregates' inherent achirality and the absence of external biases, underscoring the spontaneous nature of the mirror symmetry breaking.'

Supplementary Information: 'To determine the selection of handedness of the periodic double-twist configuration, we solve the Beris-Edwards equation² using the hybrid lattice Boltzmann method on a D3Q19 lattice¹¹. We confine the liquid crystal to a box with dimensions $[L_x, L_y, L_z]=[151, 151, 51]$ and impose finite planar surface anchoring on both walls. We use periodic boundary conditions in the x - and y -directions, and we set $A_0 = 0.05$, $L_1 = 0.05$, $L_2 = 0$, $L_3 = 0.1624$, and $L_4 = 0.0667$, corresponding to $K_1 = 3K_2 = \frac{1}{3}K_3 = K_{24}$, and flow-tumbling characteristics ($\xi = 0.6$) to describe the nematic DSCG solutions^{9, 12}. Upon application of a pressure-driven flow along the x -direction, the periodic double-twist configuration emerges robustly, irrespective of whether we start the simulations from a random initial condition or from a uniform initial condition. We find that the periodic double-twist configuration has the same handedness along the x - and z -directions, as shown in the snapshot in Fig. S7 and in Table S1). The same handedness is selected in both twist directions as it leads to a smooth variation of the twisting and avoids additional energetically costly splay deformations that would be needed for opposite-handedness of the twist. Left- and right-handedness are stochastically equal. In 21 independent simulations, we find 10 left-handed and 11 right-handed configurations, as reported in Table S1.'

Figure R1.1. Handedness of the flow-induced periodic double-twist configuration, assessed in simulations. The director field of the periodic double-twist configuration exhibits the same handedness in both the x - and z -directions (right-handed chirality in this specific example).

Table R1.1: The handedness of the periodic double-twist configuration, obtained in 21 independent simulations.

	left-handed twist in x -direction	right-handed twist in x -direction
left-handed twist in z -direction	10	0
right-handed twist in z -direction	0	11

Overall, I think the work would benefit from some further elaboration and results on the exact mechanism(s) that control the spontaneous symmetry breaking and their consequences. Also, can these mechanisms be biased to some handedness?

We appreciate the reviewer asking for a more detailed discussion on the mechanisms governing the spontaneous mirror symmetry breaking. Drawing from our simulation results, we have substantiated the framework outlined in the manuscript. Specifically, the emergence of spontaneous mirror symmetry breaking is attributed to two primary factors: the flow-tumbling character inherent to nematic DSCG solutions, and the elastic instability arising from a specific flow-induced biaxial-splay configuration. This configuration evolves towards the energetically most favorable state: the periodic double-twist configuration, as shown in Fig. R1.2.

We further outline the derivation of the Oseen-Frank elastic free energy density of the biaxial-splay, double-splay and double-twist configurations in the Supplementary Information (Eqs. S10, S11 and S12) in cylindrical coordinates. The biaxial-splay configuration has the highest energy density, as shown in Fig. R1.3, and is unstable to small perturbations, relaxing into the lowest energy density configuration, the periodic double-twist configuration.

Regarding the handedness, our simulations consistently indicate an unbiased selection. As outlined in Table R1.1, left- and right-handed configurations appear with equal probability across 21 independent simulations.

We have expanded the discussion in the Supplementary Information.

The texts now read:

‘Given that $K_3 = 3K_1 = 30K_2^9$,¹⁰, $f_{\text{biaxial-splay}}$ is larger than $f_{\text{double-splay}}$ and $f_{\text{double-twist}}$, as shown in Fig. S6. The double-twist configuration is the energetically most favorable configuration and will be selected.’

Figure R1.2. Simulation snapshots of the director field showing the time evolution from a biaxial-splay configuration (left) to the periodic double-twist configuration (right) in pressure-driven flow. Top row: top view of the center plane of the channel. Bottom row: side view of the channel. The grey rods denote the director field; the color indicates the scalar order parameter S .

Figure R1.3. Oseen-Frank elastic free energy densities of the three director configurations. The Oseen-Frank elastic free energy density of the double-twist configuration is lower than those of the biaxial-splay and double-splay configurations.

Finally, I think this is a relevant and well executed work which is appropriate and of relevance to be published in Nature Comm, upon addressing the comments above.

We are grateful for the positive feedback and the constructive comments that have helped us improve our manuscript.

References:

- [1] Zhang, R., Roberts, T., Aranson, I. S., & De Pablo, J. J. Lattice Boltzmann simulation of asymmetric flow in nematic liquid crystals with finite anchoring. *Chem. Phys.*, **144**, 084905 (2016).
- [2] Wang, W., & Zhang, R. Interplay of active stress and driven flow in self-assembled, tumbling active nematics. *Crystals*, **11**(9), 1071. (2021)

Responses to reviewers' comments

Reviewer 2

The manuscript presents an experimental liquid-crystalline system which induces a chiral symmetry breaking in an achiral material with flow.

The material in question is a lyotropic chromonic liquid crystal - they are known to promote chiral symmetry breaking due to low twist elastic constant, which has been demonstrated in various static systems before. In flow, additional torques act on the liquid crystal constituents, leading to stabilization of structures that would otherwise relax to a static ground state. The main contribution of the manuscript is demonstrating in experiment, the first flow realization of a chiral striped pattern in an achiral system. The pitch of the stripes can be regulated with flow velocity. The structure is fully backed by theoretical considerations, including stability analysis that explains the formation of the striped pattern, and relations that collapse the measurements onto the same curve. I find the analytical explanations elegant and effective. The experiments are well documented and explore the system with different techniques to provide a full picture, including a fluorescent confocal imaging.

The research is of sufficient prominence and novelty to merit publication in suggested journal. It is of interest to readers in soft matter, optics and hydrodynamics. It extends the range of materials for which microfluidic response is explored, and provides a way of generating optically interesting patterns with minimal effort.

We thank the reviewer for the careful assessment of our manuscript.

Before publication, the following should be clarified: it is not immediately clear from the text whether the chirality can alternate left and right-handed already within the same periodic pattern, or only in different realizations of the same pattern.

We thank the reviewer for raising the question about the handedness of the pattern. As it is difficult to identify the handedness of the periodic double-twist configuration in our experiments, we turn to numerical simulations using the hybrid lattice Boltzmann method to access the handedness of stripe domains [1]. To closely mirror the properties of nematic DSCG solutions [2], we implement distinct elastic constants, emphasizing a small K_2 and a large K_{24} , and ensure a flow-tumbling character. With the liquid crystal solution confined between two plates (in the z -direction), and employing no-slip and finite planar surface anchoring conditions, periodic double-twist configurations emerge upon the onset of a pressure-driven flow and small perturbations, as shown in Fig. R2.1.

Right- and left-handed chiralities emerge with equal probability in 21 independent simulations (10 left-handed and 11 right-handed configurations), and we do not observe the chirality to alternate in a given realization of the periodic double-twist configuration.

Moreover, in all 21 independent simulations, we observe that the twist deformations in the x - and z -directions exhibit the same handedness (Fig. R2.1 and Table R2.1). This can be understood by considering that opposite-handedness configurations would necessitate energetically costly splay deformations in addition to the twist deformations present in both scenarios. We have added these results in both the manuscript and the Supplementary Information.

The texts now read:

Manuscript: *‘We determine the handedness of the periodic double-twist configuration using hybrid lattice Boltzmann simulations. The twist deformations in the x - and z -directions exhibit the same handedness (see Fig. S7 and Table S1), which can be understood by considering that opposite-handedness configurations would necessitate energetically costly splay deformations in addition to the twist deformations. Across 21 independent simulations, we find an unbiased selection between left- and right-handed configurations, as outlined in Table S1. This is a reflection of the DSCG aggregates' inherent achirality and the absence of external biases, underscoring the spontaneous nature of the mirror symmetry breaking.’*

Supplementary Information: *‘To determine the selection of handedness of the periodic double-twist configuration, we solve the Beris-Edwards equation² using the hybrid lattice Boltzmann method on a D3Q19 lattice¹¹. We confine the liquid crystal to a box with dimensions $[L_x, L_y, L_z]=[151, 151, 51]$ and impose finite planar surface anchoring on both walls. We use periodic boundary conditions in the x - and y -directions, and we set $A_0 = 0.05$, $L_1 = 0.05$, $L_2 = 0$, $L_3 = 0.1624$, and $L_4 = 0.0667$, corresponding to $K_1 = 3K_2 = \frac{1}{3}K_3 = K_{24}$, and flow-tumbling characteristics ($\zeta = 0.6$) to describe nematic DSCG solutions^{9, 12}. Upon application of a pressure-driven flow along the x -direction, the periodic double-twist configuration emerges robustly, irrespective of whether we start the simulations from a random initial condition or from a uniform initial condition. We find that the periodic double-twist configuration has the same handedness along the x - and z -directions, as shown in the snapshot in Fig. S7 and in Table S1). The same handedness is selected in both twist directions as it leads to a smooth variation of the twisting and avoids additional energetically costly splay deformations that would be needed for opposite-handedness of the twist. Left- and right-handedness are stochastically equal. In 21 independent simulations, we find ten left-handed and 11 right-handed configurations, as reported in Table S1.’*

Figure R2.1. Handedness of the flow-induced periodic double-twist configuration, assessed in simulations. The director field of the periodic double-twist configuration exhibits the same handedness in both the x - and z - directions (right-handed chirality in this specific example).

Table R2.1: The handedness of the periodic double-twist configuration, obtained in 21 independent simulations.

	left-handed twist in x -direction	right-handed twist in x -direction
left-handed twist in z -direction	10	0
right-handed twist in z -direction	0	11

Additionally, chirality itself only comes to prominence in phenomena that can exploit and detect it. This should be discussed - for example, chirality of the system could be more convincingly proven by optical discrimination through different responses to circularly polarized light.

This is an interesting comment. We agree that it would be revealing to optically discriminate the chirality. However, determining the handedness of the periodic double-twist configuration in the flow direction is challenging experimentally. We have a number of exciting avenues in mind where the chirality could be exploited; for example, we envision using the stripe patterns as programmable scaffolds designed to transmit or detect chirality at the molecular level. These are longer term plans though that go beyond the scope of the current manuscript.

References:

- [1] Zhang, R., Roberts, T., Aranson, I. S., & De Pablo, J. J. Lattice Boltzmann simulation of asymmetric flow in nematic liquid crystals with finite anchoring. *Chem. Phys.*, **144**, 084905 (2016).
- [2] Wang, W., & Zhang, R. Interplay of active stress and driven flow in self-assembled, tumbling active nematics. *Crystals*, **11**(9), 1071. (2021)

Responses to reviewers' comments

Reviewer 3

Zhang and coauthors observe interesting symmetry breaking effects in driven chromonic liquid crystals. They find that the combined large elastic anisotropy of chromonics and the pumped viscous effects create beautiful stripe patterns. Unfortunately there are several glaring issues with the work as reported here that should be addressed before further consideration, both experimental and simulation/analysis related.

We are grateful for the interesting and comprehensive feedback on our manuscript.

The authors repeatedly state they used pressure control to drive the flow but the methods say they used a syringe pump—in this case, they used volume control (syringe pump) at fairly low flow rates to induce the flow. The stepping action of the syringe pump could cause issues. Would results look different with actual pressure control? The simulations seem to indicate they used pressure control. This could have substantial effects on the observed patterns.

We thank the reviewer for highlighting the important point regarding the method of flow control and the possible effect of the syringe pump's stepping action on our experimental results. Our setup employs a syringe pump that controls the volumetric flow rate, but the resulting flow is pressure-driven (but not pressure-controlled, as the reviewer rightfully points out).

To dampen any possible effects of the stepping disturbances of the syringe pump, we added a large reservoir to the microfluidic cell. This reservoir helps smooth out the flow, minimizing the pulsatile effects inherent to syringe pumps, and ensures consistent flow conditions. We further assessed the robustness of our setup by testing different microfluidic cells and syringe pumps, where some of the cells did not have a reservoir. In all scenarios, the stripe patterns consistently appear within the same range of volumetric flow rates and with the same characteristics. This gives us confidence that the stepping action of the syringe pump does not introduce variability in our results.

Regarding the interesting question about the potential differences between pressure control and volumetric flow rate control, the agreement between our numerical and experimental results indicates that the mode of flow control does not influence the emergence of the chiral structures. The choice of employing volumetric flow rate control in the experiments is primarily due to its ease of implementation. We have revised our manuscript to address the role of the stepping action of the syringe pump.

The texts now read:

‘To avoid potential pulsatile effects inherent to syringe pump-controlled flows and to ascertain a uniform velocity profile at the inlet, we connect a reservoir of 25 mm in height, 40 mm in width, and 2-3 mm in interior thickness to one end of the microfluidic cell.’

‘We ensure the robustness of our findings by conducting tests utilising different microfluidic cells and syringe pumps, including cells without a reservoir. The stripe patterns consistently form for the same range of flow velocities, demonstrating that the stepping action of the syringe pump does not affect the results.’

Reference 44 does not really support the authors’ claim regarding polar anchoring energy as it is mainly about azimuthal anchoring. (They use W_ϕ not W_θ) The true polar anchoring energy on a bare or abrasive treated glass surface is unknown but seems like it would be much higher than the authors indicate here at $1\text{e-}6\text{--}1\text{e-}7 \text{ J m}^{-2}$. Small tilts would induce large splay deformations which are especially energetically costly for the large aggregates in DSCG. This point is key to the discussion regarding saddle-splay and more clarification would help here.

We are thankful to the reviewer for pointing out our oversight regarding the reference to the surface anchoring energies in our manuscript. We agree that reference 44, which discusses the azimuthal surface anchoring strength, does not appropriately support our assertions regarding the polar surface anchoring strength.

Acknowledging the gap in the literature regarding direct measurements of the polar surface anchoring strength on treated glass surfaces, we gain further insight from a study investigating the bacteria *Bacillus subtilis* in a lyotropic chromonic liquid crystal solution in a cell with homeotropic alignment [1]. The research demonstrates that the bacteria can navigate in the direction perpendicular to the imposed homeotropic director field, indicating that the homeotropic surface anchoring is weak. The authors employ numerical simulations to estimate the homeotropic surface anchoring strength on the bacteria surface that allows the bacteria to align perpendicularly to the direction of the LCLC, and arrive at a value of $W_p \approx 10^{-6} \text{ J m}^{-2}$. Given that the anchoring condition on the bacterial surface is homeotropic, we consider this value as a reference for the polar surface anchoring strength in our system; $W_\theta \approx 10^{-6} \text{ J m}^{-2}$. This value is approximately 3.5 times larger than the measured azimuthal surface anchoring strength of $W_\phi \approx 3 \times 10^{-7} \text{ J m}^{-2}$, and can still be considered as a weak surface anchoring.

Upon revisiting our numerical analysis with this refined perspective, we find that our previously investigated range of surface anchoring strengths is still in agreement with this updated estimation. Specifically, our initial range spans from $3 \times 10^{-7} \text{ J m}^{-2}$ to $3 \times 10^{-6} \text{ J m}^{-2}$, which includes the value derived from the bacteria study. The core conclusions drawn in our manuscript regarding the saddle-splay elasticity remain thus valid under this revised assumption of the polar surface anchoring strength.

We have modified the manuscript and the Supplementary Information to specify that we discuss the ‘polar surface anchoring strength’ and we have updated the citation.

The text now reads:

Manuscript: *‘The polar surface anchoring strength of DSCG solutions, however, has been reported to be weak, on the order of 10^{-6} J m^{-2} ⁴⁴ (see Supplementary Information).’*

Supplementary Information: *‘For θ obtained in the regime of $Er_{av} = 25\text{-}50$, W is on the order of 10^{-6} J m^{-2} , as shown in Fig. S4(a). Experimentally, the polar surface anchoring strength of DSCG solutions on rubbed glass surfaces has not yet been determined. To make sure our values are in the correct range, we consider work investigating the bacteria *Bacillus subtilis* in a lyotropic chromonic liquid crystal solution in a cell with homeotropic alignment⁸. The bacteria are observed to navigate in the direction perpendicular to the imposed homeotropic director field, which indicates that the homeotropic surface anchoring is weak. The authors employ numerical simulations to estimate the homeotropic surface anchoring strength on the bacteria surface that allows the bacteria to align perpendicularly to the direction of the LCLC, and find a value of 10^{-6} J m^{-2} . This value is in agreement with the values we find by assuming $\theta_b = \theta_c$ for the boundary condition used to solve Eq. S6, which justifies our numerical procedure.’*

Do the authors see the same result on bare (not rubbed) glass? Glass treated for homeotropic alignment?

This is an interesting question, and it highlights an essential aspect of our findings - the important role of the initial surface anchoring conditions in the emergence of the periodic double-twist configuration.

We find that the stripe patterns do not emerge on bare glass and for planar surface anchoring conditions where the director is perpendicular to both the flow direction (x -direction in the manuscript) and the gap thickness direction (z -direction in the manuscript), referred to as case 2 here. This indicates that the planar surface anchoring condition with the director parallel to the flow direction used in our work (referred to as case 1) is integral to the formation of the stripe patterns. Specifically, in the range of flow velocities where the stripe patterns emerge in case 1, the director field in case 2 exhibits a pure log-rolling state. The log-rolling state persists for a large range of flow velocities, until it eventually evolves, at much higher flow velocities compared to those studied in our manuscript, into different textures. We have not yet conducted experiments using homeotropic surface anchoring conditions, but it will be interesting to do so in future work.

We have added a discussion in the manuscript, emphasizing that the formation of stripe patterns necessitates specific surface anchoring conditions.

The updated text reads:

'The biaxial-splay configuration that is the precursor for the periodic double-twist configuration emerges because of the imposed surface anchoring parallel to the flow direction. The stripe patterns do not form if we modify the surface anchoring condition so the director is planar and aligned in the y-direction; under these conditions we observe a stable log-rolling state for a comparable range of flow velocities.'

In general, the observations here remind of the Williams domain and flexo domain effects (which I believe may break mirror symmetry too) and other similar patterns that emerge in LC dynamic flows. It would be good for the authors to first revisit that deep literature. In that context, are the observations here that novel?

We thank the reviewer for allowing us to expand on the connections of our work with work reported in the literature. Here we elaborate on the unique features of our periodic double-twist structure and on the distinctions between our study and established concepts.

Williams domains form due to the rotation of the director in regions with antiparallel dipole moments, in response to an external electric field perpendicular to the internal polarization direction [2]. In these domains, opposing rotations of neighboring director fields create a splay deformation, giving rise to a 'zig-zag' configuration that appears as a stripe pattern when observed through crossed polarizers. Later work showed this mechanism to emerge in hydrodynamic instabilities in nematic liquid crystals under oscillatory shear [3-4]. When the initial surface anchoring is parallel to the shear direction, the bulk director's deviation from the surface anchored state can induce a twist deformation [3-4]. This twist deformation, however, is restricted to the gap thickness direction only. In our study, the stripe patterns originate from a periodic double-twist structure: the twist deformation is not limited to the gap thickness direction, but the director field also twists periodically in the flow direction, which results in the well-defined characteristic period of the stripes. Another important distinction lies in the initiating factors; the mirror symmetry breaking reported in Williams domains is triggered by oscillatory shear, which results in a net zero flow and introduces an asymmetry via the oscillatory shear. Our patterns emerge in a continuous shear flow with a mirror-symmetric velocity profile.

The flexo-electric effect arises from a linear coupling between an electric field and a liquid crystal distortion [5-7] and has been reported to facilitate mirror symmetry breaking under specific conditions such as asymmetric boundary conditions [7-8]. This effect gives rise to periodic structures, but it primarily induces splay and bend deformations, rather than twist deformations as those seen in our patterns [9]. The underlying mechanism of the flexo-electric effect is rooted in electric field interactions and is distinct from the flow-induced phenomenon reported in our manuscript. The initiation of our periodic chiral structure stems from instabilities driven by the saddle-splay term, showcasing a different mechanism.

The stripe patterns occurring in flowing liquid-crystalline polymers that exhibit a similar orientation perpendicular to both the flow and the thickness directions as ours [10] are characterized by a ‘zig-zag’ director field dominated by splay deformations, a configuration that differs from our findings. Additionally, these stripes are transient textures that arise during initiation or cessation of shear, while our structures form in steady-state flows.

While there are parallels with other structures emerging in flowing liquid crystals, our research illuminates three novel findings: (i) The emergence of a periodic double-twist configuration; (ii) a distinct pathway to mirror symmetry breaking using continuous flow; and (iii) the first observation of mirror symmetry breaking in flowing LCLCs.

We have added discussions to the manuscript highlighting related work:

‘Hydrodynamic instabilities induced by oscillatory shear with a zero net flow can trigger chiral Williams domains in nematic thermotropic liquid crystals^{20,21}.’

‘Mirror symmetry breaking has further been observed in Williams domains forming under oscillatory shear, where the twist deformations are in the cell thickness direction^{20,21}. Here we show that steady flow can induce and stabilise chiral structures that extend periodically in the flow direction.’

Minor points:

The authors have missed what seems like a low hanging fruit: compare energy densities in the system graphically. Elastic energies and stress energies would be revealing at a glance to the energy competition that creates the symmetry breaking. I suspect the biaxial-splay (or saddle splay concentrated) points would stand out, no?

We thank the reviewer for the nice suggestion. We revisit our derivation of the Oseen-Frank elastic free energy density in the Supplementary Information (Eqs. S10, S11 and S12) to graphically illustrate the biaxial-splay, double-splay and double-twist configurations, as shown in Fig. R3.1(a). The biaxial-splay configuration has the highest Oseen-Frank elastic free energy density, the double-twist configuration the lowest one.

In our updated hybrid lattice Boltzmann simulations, we further evaluate the prevalence of twist deformations that lead to mirror symmetry breaking upon flow application. In particular, we calculate the temporal evolution of the powers of splay, twist, bend and saddle-splay, expressed as

$$P_{splay} = \int_{\Lambda} (\nabla \cdot \mathbf{n})^2 d\Lambda, P_{twist} = \int_{\Lambda} (\mathbf{n} \cdot (\nabla \times \mathbf{n}))^2 d\Lambda, P_{bend} = \int_{\Lambda} (\mathbf{n} \times (\nabla \times \mathbf{n}))^2 d\Lambda, P_{saddle-splay} = \int_{\Lambda} -\nabla \cdot (\mathbf{n}(\nabla \cdot \mathbf{n}) + \mathbf{n} \times (\nabla \times \mathbf{n})) d\Lambda,$$

where Λ is a control volume (see Fig. R3.1(b)). Prior to the occurrence of mirror symmetry breaking, P_{twist} and $P_{saddle-splay}$ are comparable. As the double-twist configuration emerges, P_{twist} increases and becomes an order of magnitude larger than $P_{saddle-splay}$. This confirms that the twist mode is energetically favorable.

These results indicate that while both saddle-splay and twist deformations are initially present, the double-twist configuration prevails due to its lowest energy cost.

Figure. R3.1. Oseen-Frank elastic free energy densities and elastic powers of different deformations. (a) The Oseen-Frank elastic free energy density of biaxial-splay, double-twist, and double-splay configurations. (b) The elastic powers of splay, twist, bend, and saddle-splay deformations.

We have expanded the discussion in the manuscript and the Supplementary Information. The texts now read:

Manuscript: *‘We further rationalize the selection of the double-twist configuration by examining the elastic powers of splay, twist, and bend deformations in hybrid Lattice-Boltzmann simulations. The elastic power of twist is indeed much larger than the elastic powers of splay and bend (see Supplementary Information and Fig. S7).’*

Supplementary Information: ‘Given that $K_{24} = K_3 = 3K_1 = 30K_2^9,^{10}$, $f_{biaxial-splay}$ is larger than $f_{double-splay}$ and $f_{double-twist}$, as shown in Fig. S6. The double-twist configuration is the energetically most favorable configuration and will be selected.’

‘From the hybrid lattice Boltzmann simulations, we calculate the temporal evolution of the elastic powers of splay, twist, bend and saddle-splay deformations, denoted as $P_{splay} = \int_{\Lambda} (\nabla \cdot \mathbf{n})^2 d\Lambda$,

$$P_{twist} = \int_{\Lambda} (\mathbf{n} \cdot (\nabla \times \mathbf{n}))^2 d\Lambda, P_{bend} = \int_{\Lambda} (\mathbf{n} \times (\nabla \times \mathbf{n}))^2 d\Lambda, P_{saddle-splay} = \int_{\Lambda} -\nabla \cdot (\mathbf{n}(\nabla \cdot \mathbf{n}) + \mathbf{n} \times (\nabla \times \mathbf{n})) d\Lambda,$$

where Λ is a control volume (Fig. S8). Prior to the occurrence of mirror symmetry breaking, P_{twist} and $P_{saddle-splay}$ are comparable. As the double-twist configuration emerges, P_{twist} increases and becomes an order of magnitude larger than $P_{saddle-splay}$. This confirms that the twist mode is energetically favorable.’

And in the abstract, the terminology vertical to the flow direction is confusing.

We appreciate the reviewer's feedback on the terminology. We have modified this sentence in the abstract.

References:

- [1] Zhou, S., Tovkach, O., Golovaty, D., Sokolov, A., Aranson, I. S., & Lavrentovich, O. D. Dynamic states of swimming bacteria in a nematic liquid crystal cell with homeotropic alignment. *New J. Phys.*, **19**(5), 055006. (2017).
- [2] Williams, R. Domains in liquid crystals. *Chem. Phys.*, **39**(2), 384-388. (1963).
- [3] Clark, M. G., Saunders, F. C., Shanks, I. A., & Leslie, F. M. A study of flow alignment instability during rectilinear oscillatory shear of nematics. *Mol. Cryst. Liq. Cryst.*, **70**(1), 195-222. (1981).
- [4] Mullin, T., & Peacock, T. Hydrodynamic instabilities in nematic liquid crystals under oscillatory shear. *Proc. Math. Phys. Eng. Sci.*, **455**(1987), 2635-2653. (1999).
- [5] Blinov, L. M. Domain instabilities in liquid crystals. *J. Phys. Colloq.*, **40**(C3), C3-247. (1979).
- [6] Hinov, H. P., Bivas, I., Mitov, M. D., Shoumarov, K., & Marinov, Y. A further experimental study of parallel surface-induced flexoelectric domains (PSIFED) (flexo-dielectric walls). *Liq. Cryst.*, **30**(11), 1293-1317. (2003).
- [7] Palto, S. P. Dynamic and photonic properties of field-induced gratings in flexoelectric LC layers. *Crystals*, **11**(8), 894. (2021).
- [8] Zhou, X., Jiang, Y., Qin, G., Xu, X., & Yang, D. K. Static and dynamic properties of hybridly aligned flexoelectric in-plane-switching liquid-crystal display. *Phys. Rev. Appl.*, **8**(5), 054033. (2017).
- [9] Meyer, R. B. Piezoelectric effects in liquid crystals. *Phys. Rev. Lett.*, **22**(18), 918. (1969).
- [10] Larson, R. G., & Mead, D. W. Development of orientation and texture during shearing of liquid-crystalline polymers. *Liq. Cryst.*, **12**(5), 751-768. (1992).

REVIEWERS' COMMENTS

Reviewer #1 (Remarks to the Author):

The authors have well addressed my comments. I recommend publication.

Reviewer #2 (Remarks to the Author):

My main questions in my previous review were concerning the presence of domains of opposite handedness after the symmetry breaking. In their revised manuscript and in particular, the supplemental information, the authors addressed my concerns well. I now endorse the publication of this manuscript in its current form.

Reviewer #3 (Remarks to the Author):

Zhang and coauthors revised manuscript makes important clarifications and corrections. I think this work is of interest to the (sub) field and suitable for the NatComms audience. However, I am not entirely convinced by the authors' arguments regarding alignment energy comparisons to homeotropic bacteria and I have found another work observing striped patterns related to flow and anchoring conditions in chromonics:

Jeong, Joonwoo, et al. "Homeotropic alignment of lyotropic chromonic liquid crystals using noncovalent interactions." *Langmuir* 30.10 (2014): 2914-2920.

I think this work should continue to publication but I believe the authors should more clearly leave open the possibility that the mechanism for the patterns may have some other cause and address similarities to the reference mentioned above.

Responses to reviewers' comments

Reviewers 1 and 2

We are grateful to reviewers 1 and 2 for reassessing our manuscript and for recommending it for publication.

Reviewer 3

Zhang and coauthors revised manuscript makes important clarifications and corrections. I think this work is of interest to the (sub) field and suitable for the NatComms audience.

We thank the reviewer for acknowledging the clarifications made in the revised manuscript, and for recognizing the interest of the work for the readership of Nature Communications.

However, I am not entirely convinced by the authors' arguments regarding alignment energy comparisons to homeotropic bacteria and I have found another work observing striped patterns related to flow and anchoring conditions in chromonics: Jeong, Joonwoo, *et al.* "Homeotropic alignment of lyotropic chromonic liquid crystals using noncovalent interactions." *Langmuir* 30, 10 (2014): 2914-2920.

I think this work should continue to publication, but I believe the authors should more clearly leave open the possibility that the mechanism for the patterns may have some other cause and address similarities to the reference mentioned above.

We appreciate the opportunity to clarify the aspect of 'alignment energy comparisons to homeotropic bacteria'. Given the absence of direct measurements of the polar surface anchoring strength on treated glass surfaces, we refer to research on the rotational behavior of the bacteria *Bacillus subtilis* in a homeotropically aligned lyotropic chromonic liquid crystal solution. Initially, the bacteria are aligned parallel to the cell's aligned state, with the director on the bacteria surface being planarly aligned and parallel to the long axis of the bacteria. As the bacteria rotate and deviate from this initial state, the director field undergoes a splay deformation extending from the bacteria surface to the far field. This scenario involves a competition between the polar anchoring strength on the bacteria surface and the torque exerted by the bacteria flagella motors. We take the estimated value of the polar surface anchoring strength ($W_p \approx 10^{-6} \text{ J m}^{-2}$) for this system as a reference value in our study. We acknowledge that this comparison may be farfetched as the two systems are quite distinct, but it is the only way to get at least an estimate and is thus valuable information. We welcome future work of direct measurements of the polar surface anchoring strength to enhance the understanding. We have now clarified this point in the Supplementary Information, ensuring that the information is accessible and clear.

The text now reads:

‘Experimentally, the polar surface anchoring strength of DSCG solutions on rubbed glass surfaces has not yet been determined. To ensure our estimated values are within a realistic range, we draw an analogy with research on Bacillus subtilis in a lyotropic chromonic liquid crystal (LCLC) solution in a cell with homeotropic alignment⁸. In this study, the initial alignment of the bacteria is parallel to the director field, where the director on the bacteria’s surface is planarly aligned and parallel to the long axis of the bacteria. As the bacteria rotate and deviate from this initial state, the director field around the bacteria exhibits a splay deformation, extending from the bacteria surface to the far field. This scenario involves a competition between the polar anchoring strength on the bacteria surface and the torque exerted by the bacteria flagella motors. The study employs numerical simulations to estimate the polar anchoring strength on the bacteria surface allowing the bacteria to be perpendicular to the LCLC solutions director field, and reports a value of 10^{-6} J m^{-2} . This value is in agreement with our values obtained by assuming $\theta_b = \theta_c$ as the boundary condition for solving Eq. S6. This provides justification for our numerical procedure. However, it is important to note that the value of surface anchoring strength derived in the study of Bacillus subtilis may not exactly represent the polar anchoring strength on rubbed glass surfaces, and we acknowledge that direct measurements of the polar anchoring strength on rubbed glass surfaces will provide important further insight.’

The work by Jeong *et al.* (2014) is a beautiful example of another type of stripe pattern that emerges in nematic lyotropic chromonic liquid crystal solutions as a result of the small twist Frank elastic constant of LCLC solutions. The pattern occurs as the director field relaxes from a planar alignment to the homeotropic alignment imposed by the surface anchoring conditions. During the relaxation, the material undergoes twist deformations that result in stripe patterns. Similar to our periodic double-twist structure, a key ingredient for the pattern formation is the ease to undergo twist deformations. The resulting structure and in particular the conditions under which it emerges are distinct though. The pattern described in Jeong *et al.* is transient and occurs in the absence of flow, while our structure is triggered and stabilized by a steady flow. In light of this comparison and to offer a more comprehensive perspective on the phenomenon of stripe patterns in LCLC solutions, we have incorporated a reference to the study by Jeong *et al.* (2014) in the manuscript. The wealth of textures that emerge in LCLC solutions is truly fascinating and suggests that we are only at the very beginning of uncovering the ordinarily rich possibilities for structuring complex fluids using flows, controlled surface conditions or transient realignments.

The text now reads:

‘In a related context, transient stripe patterns have been observed in LCLC solutions during the relaxation of the director field from a planar alignment to a homeotropic alignment imposed by the surface anchoring conditions⁵⁰. Here we show that steady flow can induce and stabilise chiral structures that extend periodically in the flow direction.’